# EFFICIENT RANDOMIZED SMOOTHING BY DENOISING WITH LEARNED SCORE FUNCTION

## ABSTRACT

The randomized smoothing with various noise distributions is a promising approach to protect classifiers from $\ell_p$ adversarial attacks. However, it requires an ensemble of classifiers trained with different noise types and magnitudes, which is computationally expensive. In this work, we present an efficient method for randomized smoothing that does not require any re-training of classifiers. We built upon denoised smoothing, which prepends denoiser to the pre-trained classifier. We investigate two approaches to the image denoising problem for randomized smoothing and show that using the score function suits for both. Moreover, we present an efficient algorithm that can scale to randomized smoothing and can be applied regardless of noise types or levels. To validate, we demonstrate the effectiveness of our methods through extensive experiments on CIFAR-10 and ImageNet, under various $\ell_p$ adversaries.

## 1 INTRODUCTION

The deep image classifiers are susceptible to deliberate noises as known as *adversarial attacks* (Szegedy et al., 2013; Goodfellow et al., 2014; Carlini & Wagner, 2017). Even though many works proposed heuristics that can annul or mitigate adversarial attacks, most of them were broken by stronger attacks (Athalye et al., 2018; Athalye & Carlini, 2018). The vulnerability of empirical defenses had led the researchers to scrutinize on *certified defenses*, which ensure the models to have constant output within the allowed set around given input. Unfortunately, many provable defenses are not feasible to large-scale neural networks because of their constraints on the architecture.

On the other hand, randomized smoothing is a practical method that does not restrain the choice of neural networks. The randomized smoothing converts any base classifier to a smoothed classifier by making predictions over randomly perturbed samples. Then the smoothed classifiers are guaranteed to have a $\ell_p$ certified radius, which is theoretically derived by the noise type used for smoothing. Since Cohen et al. (2019) derived tight $\ell_2$ certified radius for Gaussian randomized smoothing, sequential works studied the certification bounds for various distributions (Teng et al., 2020; Yang et al., 2020). As base classifiers are required to predict randomly perturbed samples, natural classifiers are not sufficient for randomized smoothing. Therefore, many works proposed training ensemble of base classifiers accustomed for randomized smoothing. However, since each trained classifier only applies to specific noise distribution and level, it is expensive to protect against various $\ell_p$ adversaries and robustness strength.

In this work, we tackle the inefficiency of training random-ensemble of base classifiers by using one universal image denoiser to the pre-trained classifier. The idea of using denoiser for randomized smoothing was first introduced by Salman et al. (2020) and is refer to *denoised smoothing*. One step further, we study general image denoising problem for randomized smoothing with two different approaches: 1) direct training of image denoiser, and 2) solve the optimization problem by using a generative model to project to the learned data manifold. Then, we show that the score function, which is the gradient of log-density, is crucial for both approaches. We exploit multi-scale denoising score matching (Song & Ermon, 2019) for score estimation, and propose an efficient algorithm simulated annealing for image denoising. Remark that we only require one score network to certify various noise distributions and levels. We provide experimentations on ImageNet and CIFAR-10 datasets to show the efficacy of our methods. Specifically, our denoisers perform better than original denoised smoothing, while can be applied to various noise types without any re-training. Further-

more, we compare with the random-ensemble based method, which we refer to *white-box smoothing*, and show that our method works are comparable to them. In sum, we list our contributions:

- We propose novel score-based image denoisers for randomized smoothing.
- We improve denoised smoothing, which was originally proposed by Salman et al. (2020) and generalize to other distributions without training any neural networks.

## 2 RANDOMIZED SMOOTHING AND DENOISED SMOOTHING

### 2.1 BACKGROUNDS ON RANDOMIZED SMOOTHING

Let $f : \mathbb{R}^d \to \mathcal{Y}$ be a classifier and $q$ be a distribution on $\mathbb{R}^d$. Then the *randomized smoothing* with $q$ is a method that converts the base classifier $f$ to the *associated smoothed classifier* $g$, where $g(\mathbf{x})$ returns the class which is most likely to be predicted by the base classifier $f$ when $\mathbf{x}$ is perturbed by a random noise sampled from $q$, i.e.,

$$g(\mathbf{x}) = \arg\max_{c \in \mathcal{Y}} \Pr_{\mathbf{u} \sim q(\mathbf{u})} \big[ f(\mathbf{x} + \mathbf{u}) = c \big]. \tag{1}$$

The noise distribution is usually a symmetric log-concave distribution, i.e. $q(\mathbf{u}) = \exp(-\phi(\mathbf{u}))$ for some even and convex $\phi$. Note that to control the robustness/accuracy tradeoff, we embed the noise level $\lambda$ to $q$, then we have $q_\lambda(\mathbf{u}) = \exp(-\phi(\frac{\mathbf{u}}{\lambda}))$. We mix the notations $q$ and $q_\lambda$ throughout the paper.

**Robustness guarantee for smoothed classifiers** Suppose an adversary can perturb the input $\mathbf{x}$ inside the allowed set $\mathcal{B}$, which is usually an $\ell_p$ ball centered at $\mathbf{x}$. For the case when $\mathcal{B}$ is $\ell_2$ ball and $q$ is Gaussian distribution $\mathcal{N}(0, \sigma^2 I)$, $g(\mathbf{x})$ is robust within the radius

$$R = \frac{\sigma}{2} \left( \Phi^{-1}(p_1) - \Phi^{-1}(p_2) \right) \tag{2}$$

where $\Phi$ is inverse cumulative distribution function, and $p_1 = \max_c \Pr[f(\mathbf{x} + \mathbf{u}) = c]$ and $p_2 = \max_{c \neq g(\mathbf{x})} \Pr[f(\mathbf{x} + \mathbf{u}) = c]$. Cohen et al. (2019) first derived the certified radius by using Neyman-Pearson lemma, and later Salman et al. (2019a) showed alternative derivation using the Lipschitz property of smoothed classifier. Furthermore when $q$ is a centered Laplace distribution, the robustness certificate for $\ell_1$ radius was derived by Teng et al. (2020). Later, the proof methods are generalized to various distributions (may not be log-concave) that can certify various $\ell_p$ radius (Yang et al., 2020). Remark that the robustness guarantee depends on the noise distribution $q_\lambda$ and the performance of base classifier $f$ under random perturbation with $q_\lambda$.

### 2.2 RANDOMIZED SMOOTHING VIA IMAGE DENOISING

Even though the randomized smoothing can convert any classifier to a provably robust classifier, the smoothed classifier from natural classifiers are below the standard as they are not capable of predicting randomly perturbed samples. Many previous studies focused on training classifiers accustomed to randomized smoothing, which spans from noisy data augmentation (Cohen et al., 2019; Li et al., 2019) to its variants such as adversarial training (Salman et al., 2019a) or stability training (Lee et al., 2019; Zhai et al., 2019). However, such methods are computationally expensive and require a massive number of classifiers per noise types and levels.

The idea of prepending denoiser to the classifier was first introduced by Salman et al. (2020). By training denoiser $\mathcal{D}_\theta : \mathbb{R}^d \to \mathbb{R}^d$, the smoothed classifier converted from $f \circ \mathcal{D}_\theta$ outperforms 'no-denoiser' baseline. They proposed training denoisers with mean squared error (MSE) loss or classification (CLF) loss, or combining both methods. Formally, they are

$$L_{\text{MSE}}(\theta) = \mathbb{E}_{\mathbf{x} \sim p, \mathbf{u} \sim q}[\|\mathcal{D}_\theta(\mathbf{x} + \mathbf{u}) - \mathbf{x}\|^2], \tag{3}$$

$$L_{\text{CLF}}(\theta) = \mathbb{E}_{\mathbf{x} \sim p, \mathbf{u} \sim q}[\mathcal{L}_{\text{CE}}(F(\mathcal{D}_\theta(\mathbf{x} + \mathbf{u})), f(\mathbf{x}))]. \tag{4}$$

where $\mathcal{L}_{\text{CE}}$ is the cross-entropy loss and $F$ is soft version of hard classifier $f$. They showed that training with CLF loss makes perform better than denoiser with only MSE loss. Alternatively, Saremi & Srivastava (2020) trained neural empirical bayes estimator that can refine the white noise. Nonetheless, those methods still suffer from expensive training of numerous denoisers with respect to each noise types and levels.

## 3 SCORE-BASED IMAGE DENOISING

### 3.1 FORMULATION OF IMAGE DENOISING PROBLEM

The image denoising is an example of linear inverse problem, which can be formulated as following: given an observation $\mathbf{y} = \mathbf{x} + \mathbf{u}$ with $\mathbf{u} \sim q(\mathbf{u})$ finds $\hat{\mathbf{x}}(\mathbf{y})$ that is close to original $\mathbf{x}$. Let $\mathbf{x} \sim p(\mathbf{x})$ then the distribution of $\mathbf{y}$ is $p_q(\mathbf{y}) = \int p(\mathbf{y}, \mathbf{x})d\mathbf{x} = \int p(\mathbf{y}|\mathbf{x})p(\mathbf{x})d\mathbf{x} = \int q(\mathbf{y} - \mathbf{x})p(\mathbf{x})d\mathbf{x} = (p * q)(\mathbf{y})$.

**One-step denoiser** Like equation 3, the most common approach to achieve denoiser is to train denoising autoencoder (DAE) $\mathcal{D}_\theta$ with MSE loss (Zhang et al., 2017; **?**). Suppose $q$ is a Gaussian distribution $\mathcal{N}(0, \sigma^2 I)$ and let the distribution of $\mathbf{y}$ by $p_{\sigma^2}$. Then the following proposition (Robbins, 1956; Lu & Stephens, 2019; Saremi & Hyvarinen, 2020) reveals the relationship between the optimal denoiser $\mathcal{D}_{\theta^*}$ and $p_{\sigma^2}$.

**Proposition 3.1.** *Assume $\theta* \in \arg\min_\theta L_{MSE}(\theta)$, then the following equation holds:*

$$\mathcal{D}_{\theta^*}(\mathbf{y}) = \mathbf{y} + \sigma^2 \nabla_\mathbf{y} \log p_{\sigma^2}(\mathbf{y}) \tag{5}$$

The proof of proposition 3.1 is in Appendix A. Let us define the score function of density $p(\mathbf{x})$ by $\nabla_\mathbf{x} \log p(\mathbf{x})$, then the optimal DAE can be obtained by estimating the score of $p_{\sigma^2}$. Let $\mathbf{s}_\theta(\cdot; \sigma)$ be score network that estimates score of smoothed density $p_{\sigma^2}$. Then the denoiser from $\mathbf{s}_\theta$ is given by

$$\hat{\mathbf{x}}(\mathbf{y}) = \mathbf{y} + \sigma^2 \mathbf{s}_\theta(\mathbf{y}; \sigma). \tag{6}$$

Remark that it is only valid when $q$ is Gaussian distribution.

**Multi-step denoiser** Consider the maximum a posteriori (MAP) estimator that maximizes the conditional distribution $p(\mathbf{x}|\mathbf{y})$. Formally the MAP loss is given by,

$$\arg\min_\mathbf{x} L_{\text{MAP}}(\mathbf{x}; \mathbf{y}) = \arg\min_\mathbf{x} -\log p(\mathbf{x}|\mathbf{y}) \tag{7}$$

$$= \arg\min_\mathbf{x} -\log p(\mathbf{x}) - \log p(\mathbf{y}|\mathbf{x}) + \log p(\mathbf{y}) \tag{8}$$

$$= \arg\min_\mathbf{x} -\log p(\mathbf{x}) - \log q(\mathbf{y} - \mathbf{x}) \tag{9}$$

$$= \arg\min_\mathbf{x} -\log p(\mathbf{x}) + \phi(\mathbf{y} - \mathbf{x}). \tag{10}$$

Note that we simply remove density term $p(\mathbf{y})$ and rewrite with $q$. Lastly, we rewrite $q$ with $\phi$. Since the density $p(\mathbf{x})$ is usually intractable for high-dimensional dataset, one may use approximation to make the MAP loss tractable. Many recent works focused on using cutting edge generative models such as generative adversarial network (GAN) or invertible neural networks to approximate $p(\mathbf{x})$ in equation 9 (Ulyanov et al., 2018; Whang et al., 2020; Asim et al., 2020). However, GAN suffer from mode collapse, and invertible neural networks require extremely long steps to reach local minima, which are not sufficient for randomized smoothing.

Instead, we aim to approximate the gradient of $L_{\text{MAP}}$ by the score of Gaussian smooth densities. Let the approximate MAP loss with $\tilde{\sigma}$ by

$$L_{\text{MAP},\tilde{\sigma}}(\mathbf{x}; \mathbf{y}) = -\log p_{\tilde{\sigma}^2}(\mathbf{x}) + \phi(\mathbf{y} - \mathbf{x}). \tag{11}$$

Then we can approximate the gradient of $L_{\text{MAP},\tilde{\sigma}}(\mathbf{x}; \mathbf{y})$ by score network and perform gradient descent initialized with $\mathbf{x}_0 = \mathbf{y}$ as following:

$$\mathbf{x}_{t+1} = \mathbf{x}_t - \alpha \nabla_{\mathbf{x}_t} L_{\text{MAP},\tilde{\sigma}}(\mathbf{x}; \mathbf{y}) \approx \mathbf{x}_t + \alpha(\mathbf{s}_\theta(\mathbf{x}_t; \tilde{\sigma}) + \nabla_{\mathbf{x}_t} \phi(\mathbf{y} - \mathbf{x}_t)). \tag{12}$$

Remark that the proposed method can be applied to any log-concave noise distributions. Following theorem shows the recovery guarantee of our methods when $q$ is a Gaussian distribution.

**Theorem 3.2.** *Let $\mathbf{x}^*$ be local optimum of $p(\mathbf{x})$, and $\mathbf{y} = \mathbf{x}^* + \mathbf{u}$ where $\mathbf{u} \sim \mathcal{N}(0, \sigma^2 I)$. Assume $-\log p$ is $\mu$-strongly convex within the neighborhood $\mathcal{B}_r(\mathbf{x}) = \{\mathbf{z} : \|\mathbf{z} - \mathbf{x}\| \leq r\}$. Then, the gradient descent method on approximate loss $L_{MAP,\tilde{\sigma}^2}(\mathbf{x}; \mathbf{y})$ initialized by $\mathbf{x}_0 = \mathbf{y}$ converges to its local minima $\hat{\mathbf{x}}(\mathbf{y}; \tilde{\sigma}) \in \arg\min L_{MAP,\tilde{\sigma}^2}(\mathbf{x}; \mathbf{y})$ that satisfies:*

$$\mathbb{E}\|\hat{\mathbf{x}}(\mathbf{y}; \tilde{\sigma}) - \mathbf{x}^*\|_2 \leq \frac{\sigma\sqrt{d}(1 + \mu\tilde{\sigma}^2)}{1 + \mu\tilde{\sigma}^2 + \mu\sigma^2} + \tilde{\sigma}\sqrt{d} \tag{13}$$

The proof of theorem 3.2 is in Appendix A. Remark that the upper bound in equation 13 increases as $\sigma$ increases, which shows that the recovery becomes harder as $\sigma$ becomes larger. Also the upper bound is strictly increasing function of $\tilde{\sigma}$, and has the minimum when $\tilde{\sigma} = 0$.

## 3.2 EFFICIENT IMAGE DENOISING WITH SIMULATED ANNEALING

From theorem 3.2, for small $\tilde{\sigma}$ the error bound is tight but the approximation is inaccurate at nascent steps. Otherwise, when $\tilde{\sigma}$ is large, the error bound is too large. To arbiter the tradeoff, and to make the method scalable, we propose simulated annealing for score-based image denoising. Let $\{\sigma_i\}_{i=1}^{L}$ be a decreasing sequence of noise levels, then simulated annealing runs $T$ steps of approximate gradient descent for each $\sigma_i$. The algorithm for simulated annealing for image denoising is in Algorithm 1.

---

**Algorithm 1** Simulated Annealing for denoising

**Require:** $\mathbf{y}, \{\sigma_i\}_{i=1}^{L}, \alpha, T$
1: initialize $\mathbf{x}_0 = \mathbf{y}$
2: **for** $i \leftarrow 1 : L$ **do**
3:     $\alpha_i \leftarrow \alpha \cdot \sigma_i^2 / \tilde{\sigma}^2$
4:     **for** $t \leftarrow 1 : T$ **do**
5:         $\mathbf{x}_{t+1} \leftarrow \mathbf{x}_t + \alpha_i \big( \mathbf{s}_{\theta, \sigma_i}(\mathbf{x}_t) + \nabla_{\mathbf{x}_t} \phi(\mathbf{x}_t - \mathbf{y}) \big)$
6:     **end for**
7:     $\mathbf{x}_0 \leftarrow \mathbf{x}_T$
8: **end for**
9: **return** $\mathbf{x}_T$

---

Note that Song & Ermon (2019; 2020) used annealed Langevin dynamics for generative modeling. Our approach is similar to them, but we consider the image denoising problem instead. Also, note that Kingma & Cun (2010) trained score network for image denoising, but they used primitive neural networks where exact score-matching was possible.

## 3.3 SCORE ESTIMATION VIA SCORE MATCHING

Score estimation has been studied through various topics such as generative modeling (Song et al., 2020; Song & Ermon, 2019) and reinforcement learning (Sutton et al., 2000). Score matching is a method that trains a score network $\mathbf{s}_\theta(\mathbf{x})$ to estimate score. The original score matching objective is given by

$$\mathbb{E}_{\mathbf{x} \sim p(\mathbf{x})} \left[ \mathrm{tr}(\nabla_{\mathbf{x}} \mathbf{s}_\theta(\mathbf{x})) + \frac{1}{2} \|\mathbf{s}_\theta(\mathbf{x})\|_2^2 \right]. \tag{14}$$

However, due to heavy computation of $\mathrm{tr}(\nabla \mathbf{s}_\theta(\mathbf{x}))$, and since we are only interested in score of smoothed densities, we use different approach.

**Denoising Score Matching** Denoising score matching is a method that learns the score of smooth densities. More concretely, the score network $\mathbf{s}_\theta$ estimates the score of density $p_{\sigma^2}(\mathbf{y}) = \int \mathcal{N}(\mathbf{x}, \sigma^2 I) p(\mathbf{x}) d\mathbf{x}$. The objective was proved to be equivalent to the following (Vincent, 2011):

$$\mathbb{E}_{\mathbf{y} \sim q_{\sigma^2}(\mathbf{y}|\mathbf{x}), \mathbf{x} \sim p(\mathbf{x})} [\|\mathbf{s}_\theta(\mathbf{y}; \sigma) - \nabla_y \log q_{\sigma^2}(\mathbf{y}|\mathbf{x})\|_2^2]. \tag{15}$$

Remark that the optimal score network satisfies $\mathbf{s}_{\theta^*}(\mathbf{x}; \sigma) = \nabla \log p_{\sigma^2}(\mathbf{x})$ for each $\sigma$, and as $\sigma \to 0$, $\mathbf{s}_{\theta^*, \sigma}(\mathbf{x}) \to \nabla \log p(\mathbf{x})$.

**Multi-Scale Denoising Score Matching** Recently, training score network with multi-scale denoising score matching has been proposed (Song & Ermon, 2019). Multi-scale denoising score matching trains one score network with various noise magnitudes. Given a sequence of noise levels $\{\sigma_i\}_{i=1}^{L}$, which is the variance of centered Gaussian distribution, by rewriting the denoising score matching objective for each $\sigma_i$, we have

$$\mathcal{L}(\theta; \sigma_i) = \frac{1}{2} \mathbb{E}_{\mathbf{x} \sim p, \mathbf{y} \sim \mathcal{N}(\mathbf{x}, \sigma_i^2 I)} \left[ \left\| \mathbf{s}_\theta(\mathbf{y}; \sigma_i) + \frac{\mathbf{y} - \mathbf{x}}{\sigma_i^2} \right\|_2^2 \right]. \tag{16}$$

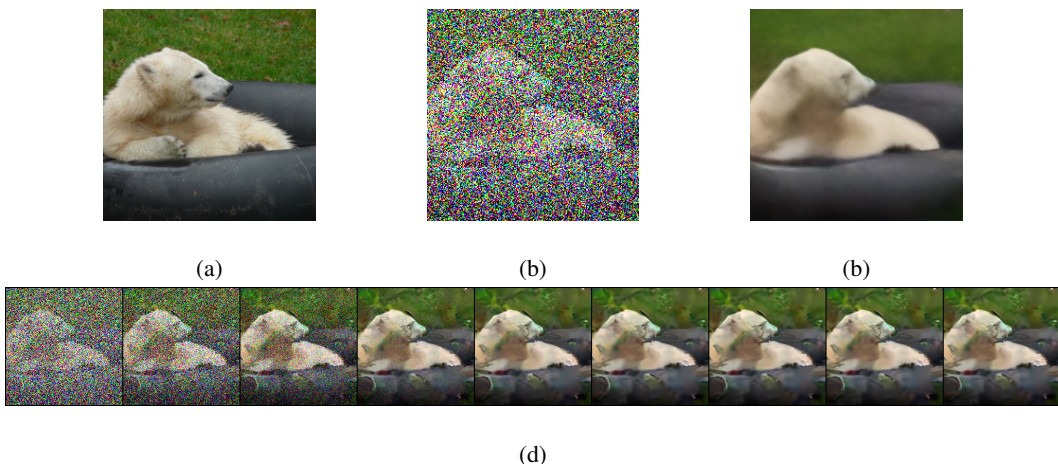

Figure 1: (a) the original image sampled from ImageNet dataset, (b) the perturbed image with Gaussian noise($\sigma = 1.0$), (c) the denoised image with one-step denoiser, (d) the progress of multi-step denoiser, the last image is the final denoised image.

Then the total loss is

$$\mathcal{L}(\theta; \{\sigma_i\}_{i=1}^{L}) = \frac{1}{L} \sum_{i=1}^{L} \sigma_i^2 \mathcal{L}(\theta; \sigma_i), \tag{17}$$

note that each loss is weighted by $\sigma_i$ which allows the loss of each noise level has the same order of magnitude. It is worth to notify that our method is unsupervised, and classifier-free.

Here we demonstrate some advantages of multi-scale denoising score matching. First, through learning various noise magnitudes at once, it suffices to train only one neural network to apply image denoising. Therefore, we can do randomized smoothing regardless of the noise level. Second, the noise makes the support of the score function to be whole space, making score estimation more consistent. Moreover, a large amount of noise fills the low-density region, which helps to estimate the score of the non-Gaussian or off-the-manifold samples. Empirically, we found out that multi-scale learning helps the denoising performance. See Appendix C for details.

## 4 EXPERIMENTS

We study the performance of our proposed denoiser applied for randomized smoothing. We experimented on ImageNet (Deng et al., 2009) and CIFAR-10 Krizhevsky et al. (2009) datasets. For comparison, we measured the certified accuracy at $R$, which is the fraction of test set for which the smoothed classifier correctly predicts and certifies robust at an $\ell_p$ radius bigger than $R$. Due to computational issue, we conducted our experiments with $N = 10,000$ samples and failure probability $\alpha = 0.001$. Everything besides, we follow the same experimental procedures as in Cohen et al. (2019). At first, we depict the perceptual performance of our proposed denoisers.

### 4.1 VISUAL PERFORMANCE OF PROPOSED DENOISERS.

We demonstrate the visual performance of our denoiser. For an image sampled from the ImageNet dataset, we perturbed the image with Gaussian noise ($\sigma = 1.0$), and the denoised images from each one-step and multi-step methods are different. Note that the result from the one-step denoiser is more blurry, but the multi-step denoiser produces a sharper edge. We refer to Appendix D for more examples of CIFAR-10 and ImageNet under various noise types.

### 4.2 CERTIFICATION WITH ONE-STEP DENOISER

We experimented the performance of one-step denoiser for Gaussian randomized smoothing. We compare with 1) *white-box smoothing*, which is canonical approach that trains base classifiers with

| $\ell_2$ radius (CIFAR-10) | 0.25 | 0.50 | 0.75 | 1.00 | 1.25 | 1.50 |
|---|---|---|---|---|---|---|
| white-box smoothing (Cohen et al., 2019) | 59 | 45 | 31 | 21 | 18 | 13 |
| denoised smoothing (Query Access) (Salman et al., 2020) | 45 | 20 | 15 | 13 | 11 | 10 |
| denoised smoothing (Full Access) (Salman et al., 2020) | 56 | 41 | 28 | 19 | 16 | 13 |
| denoised smoothing (Our method) | 60 | 42 | 28 | 19 | 11 | 6 |

Table 1: Certified accuracy of ResNet-110 on CIFAR-10 at various $\ell_2$ radii.

| $\ell_2$ radius (ImageNet) | 0.25 | 0.50 | 0.75 | 1.00 | 1.25 | 1.50 |
|---|---|---|---|---|---|---|
| white-box smoothing (Cohen et al., 2019) | 62 | 52 | 45 | 39 | 34 | 29 |
| denoised smoothing (Query Access)(Salman et al., 2020) | 48 | 31 | 19 | 12 | 7 | 4 |
| denoised smoothing (Full Access)(Salman et al., 2020) | 50 | 33 | 20 | 14 | 11 | 6 |
| denoised smoothing (Our method) | 56 | 41 | 30 | 24 | 17 | 11 |

Table 2: Certified accuracy of ResNet-50 on ImageNet at various $\ell_2$ radii.

Gaussian data augmentation (Cohen et al., 2019), and 2) the denoised smoothing with denoisers trained by Salman et al. (2020). As Salman et al. (2020) trained denoisers with various methods, we just compare with their best values. Note that they assumed query access and full access, which are discriminated based on how much information on the base classifier is provided. Remark that our method is 'no access' that we don't need any classifier information. For all experiments on denoised smoothing, we used same ResNet110 classifier for CIFAR-10 and pytorch pretrained ResNet50 classifier for ImageNet. In addition, as our method is unbiased to the base classifiers, we found out that using stronger classifier results in better certified accuracy. See Appendix C for additional experiments.

**CIFAR-10** The results for CIFAR-10 are shown in Table 1. Remark that even without using classifier loss, our method outperforms the query access baseline, and slightly better than the full access baseline. Also, the results are comparable to white-box smoothing, which is an upper bound on our framework. We suspect two reasons for the performance boost: the use of better architecture and the effect of multi-scale training. We conducted additional experiments on the effect of multi-scale training, and found out that multi-scale training helps learning the score estimation and therefore helps denoised smoothing. The results for additional experiments are in Appendix C. However, note that using classifier loss helps for large radii certification because the images denoised from large-scale noise is too blurry that conventional classifiers can't predict it.

**ImageNet** The results for ImageNet are shown in Table 2. Note that our method outperforms previous denoised smoothing baselines. We believe the same reason as in CIFAR-10. However, there is still a large gap between denoised smoothing and white-box smoothing, which is due to the difficulty of learning score function of high-resolution images.

### 4.3 Certification with multi-step denoiser

We demonstrate the effectiveness of our multi-step denoiser on denoised smoothing using various noise types. For a baseline, we compare with white-box smoothing which is training classifiers with noisy data augmentation. We experimented on Gaussian noise (Cohen et al., 2019), Laplace noise (Teng et al., 2020), and uniform noise (Yang et al., 2020) for both CIFAR-10 and ImageNet. For all experiments, we used ResNet110 classifiers for CIFAR-10 and ResNet50 classifiers for ImageNet. See Appendix B for more details. It is important to claim that all experiments are done with the only **one** score-network for each CIFAR-10 and ImageNet.

Note that for CIFAR-10, the denoised smoothing with our denoiser is slightly worse than white-box smoothing except for uniform distribution. As Yang et al. (2020) reported, the uniform distribution is well-fitted to the convolutional neural network, therefore the white-box smoothing achieves higher performance. For ImageNet, we've found out that score estimation on ImageNet is difficult, and the

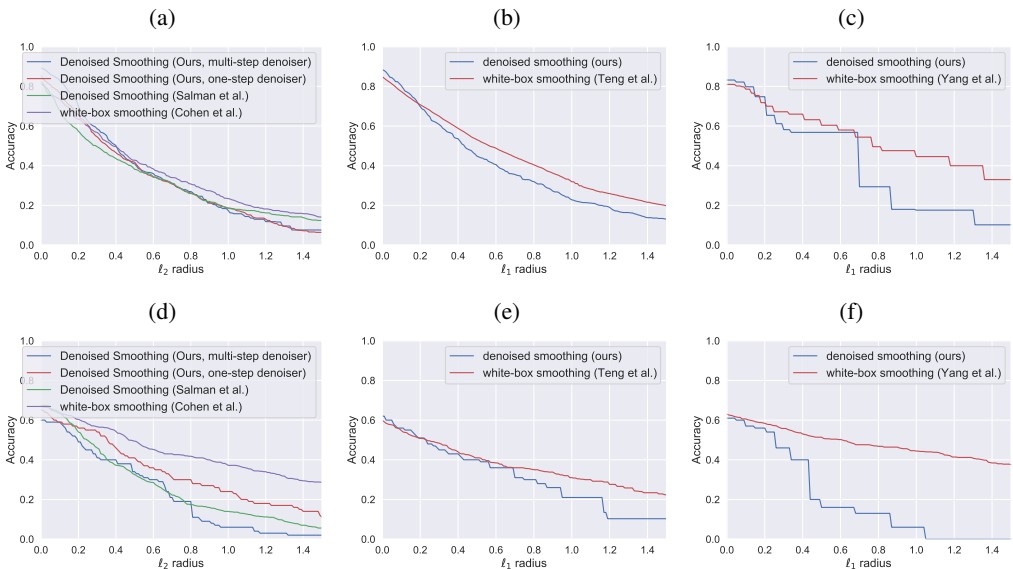

Figure 2: The performance of multi-step denoiser for denoised smoothing. The blue lines are our methods, and red lines are white-box smoothing which are experimented by each authors. (a) $\ell_2$ certified accuracy with Gaussian smoothing on CIFAR-10, (b) $\ell_1$ certified accuracy with Laplace smoothing on CIFAR-10, (c) $\ell_1$ certified accuracy with uniform smoothing on CIFAR-10, (d) $\ell_2$ certified accuracy with Gaussian smoothing on ImageNet, (e) $\ell_1$ certified accuracy with Laplace smoothing on ImageNet, (f) $\ell_1$ certified accuracy with uniform smoothingon ImageNet,.

denoising algorithm takes too long time to be certify with myriad of samples. However, we found out that our approach can stack up against white-box smoothing.

## 5  RELATED WORKS

### 5.1  DEFENSE AGAINST ADVERSARIAL ATTACKS

**Empirical Defense methods**  The empirical defenses include erasing adversarial perturbation and making models predict well in the presence of adversarial examples. The former defenses are similar to our approach that they use trained denoiser (Meng & Chen, 2017; Liao et al., 2018), or project the adversarial examples to the learned data manifold using generative models (Song et al., 2017; Samangouei et al., 2018). However, all these methods are broken by adaptive attacks (Athalye et al., 2018; Athalye & Carlini, 2018; Tramer et al., 2020), while our method has provable robustness. The latter defenses are referred to adversarial training (Madry et al., 2017; Kurakin et al., 2016; Zhang et al., 2019), which augments adversarial examples at training. Although the adversarial training methods are shown to have great empirical robustness against various adversarial attacks, they suffer from the undiscovered attacks.

**Certified Defense methods**  provides provable guarantees that the classifier's prediction remains unchanged within a neighborhood of an input. Those methods are mainly based on certification methods that are either exact or conservative. The exact certification methods are based on Satisfiability Modulo theories solvers (Katz et al., 2017; Ehlers, 2017) or mixed-integer linear programming (Fischetti & Jo, 2018; Tjeng et al., 2019; Lomuscio & Maganti, 2017). However, those methods have the computational burden and depend on the architecture of the neural network. Otherwise, conservative methods are based on Lipschitz bound of the neural network, which is more computationally efficient (Jordan et al., 2019; Salman et al., 2019b; Wong & Kolter, 2018; Raghunathan et al., 2018). However, the above methods aren't scaled for practical neural networks. Instead, randomized smoothing which is the Weierstrass transformation of a classifier is shown to be scalable with architecture independence.

**Randomized smoothing** was first presented with guarantee derived from differential privacy perspective (Lecuyer et al., 2019). Then using Li et al. (2019) showed tighter certificates using $\alpha$-divergence minimization of original and smoothed distributions. Recently, Cohen et al. (2019) proposed the tightest $\ell_2$ robust guarantee with Gaussian distribution. Furthermore, series of works derived certification bounds for various $\ell_p$ adversaries including $\ell_1$-adversary (Teng et al., 2020), $\ell_\infty$ (Zhang et al., 2020) and $\ell_0$ (Lee et al., 2019; Levine & Feizi). Later, Yang et al. (2020) showed generic proof methods for certification with Wulff Crystal theory.

Even though randomized smoothing does not constrain the base classifier, to achieve non-trivial robustness, several works have proposed custom-trained methods for randomized smoothing (Lecuyer et al., 2019; Cohen et al., 2019; Salman et al., 2019a; 2020; Yang et al., 2020). Alternatively, Lecuyer et al. (2019) trained denoising autoencoders to promote to scale PixelDP to practical neural networks. Our work is based on Salman et al. (2020), with some improvements and generalizations. Note that 1) our approach does not require any information on the base classifier, and 2) we propose general image denoising that doesn't require training denoisers per noise types or levels.

## 5.2 Image Denoising

The image denoising had a huge development by exploiting deep neural networks (Zhang et al., 2017; Jin et al., 2017; Tai et al., 2017). Moreover, inverse imaging using generative models have been studied. Ulyanov et al. (2018) showed that GAN can act as an image prior and be used for various inverse imaging problems. On the other hand, Asim et al. (2019) claimed that GAN suffers from mode collapse, and is biased toward the dataset that isn't sufficient for general image denoising. Instead, several studies (Asim et al., 2019; Whang et al., 2020) showed that invertible neural networks such as Glow (Kingma & Dhariwal, 2018) or RealNVP (Dinh et al., 2016), can be used for the deep image before various inverse imaging applications. Our work is based on score function, where using score function for inverse imaging is less studied. Note that Kingma & Cun (2010) used regularized score matching for image denoising, but their neural network is primeval, and regularized score matching is hard to be scale to practical neural networks.

## 6 Conclusion

In this work, we presented a score-based image denoising methods for randomized smoothing. Our method does not require any re-training of classifiers, and trains only one score network that can be used for denoising of any noise type and level. We empirically found out that our denoiser performs better than conventional image denoisers and denoisers trained with the classification loss, while comparable to the random ensemble approach.

We believe that current randomized smoothing is theoretically well-designed but needs to be scalable to be deployed for real world applications. On that perspective, our approach is a good initial point that can endow robustness to any classifier without any re-training. However, the hardness of estimating score function of high-dimensional data should be compromised. We believe using better architecture or devising faster optimization algorithm might help.

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
