# OpenReview forum: "Efficient randomized smoothing by denoising with learned score function"
_ICLR.cc/2021/Conference — Reject_

### Official Review · AnonReviewer1 · 2020-10-27
**I am not an expertise of the area of this paper. But I think the idea is interesting.**

**Rating:** 6
**Confidence:** 1

**Review:**

This paper proposes a method based on denoising to protect classifiers from adversarial attacks. Unlike existing methods based on randomized smoothing with various noise distributions to retrain several classifiers, the proposed one uses denoising as the preprocess of the classifier. The experimental results demonstrate the proposed method has good performance.

---

> ### Author Response · Authors · 2020-11-13
> **Response to AnnonReviewer 1**
>
> We sincerely appreciate your instructive review. Our work proposes score-based model for image denoising that can efficiently generate smoothed classifier without retraining classifiers. If there is any further question, let us know.

---

### Official Review · AnonReviewer4 · 2020-10-28
**Promising results,  clarifications would be appreciated**

**Rating:** 6
**Confidence:** 2

**Review:**

Summary & contribution:

in this paper the authors propose an improved method for randomized smoothing (for provable defense) which performs better and is less computationally expensive than previous work. More specifically the authors propose two image denoising algorithms (based on score estimation) that can be applied regardless of noise level/type. In fact, the paper is mainly an improvement of Salman et al. (2020) with a blind denoiser. Therefore, it is not needed to train different models for different kind of attacks. Another advantage of the method is that one does not need to retrain the classifier and does not even need information from the pretrained classifier while the method from Salman et al.  requires access to the classifier.


Strengths:

-strong quantitative results. Experimental section is promising. The gap with white box smoothing is small on cifar. The method outperforms Salman et al.

-Only need to train one score network to handle various types of noise type/level

-Denoiser doesn't need access to the pretrained classifier.

Weaknesses:

-In my opinion writing can be improved. I am not very familiar with the literature and it took me some time to understand section 3. More specifically I did not understand the motivation for using score based denoisers rather than a more "standard" algorithm for blind denoising.

-Method seems to be effective for low-resolution images only. The gap with white box increases on Imagenet.

Questions for the authors:

-If I understand well, one-step denoiser can only handle gaussian noise whereas multi-step can be applied to any log concave distribution? What is the advantage of one-step denoiser over multistep, does it perform better for gaussian noise?

-It is not clear to me why the method does not make use of the pretrained classifier but still outperforms Salman et al. which can acess the classifier? Is it only due to the denoiser's performance?

-I do not understand well the motivation for using score-based methods. Why scored based denoising in particular and not other blind denoising models? The current method is impacted by the size of the images, while most of the existing blind denoising algorithms are not affected by the size of the image. Actually, why even focus on learning base methods? Cannot simpler methods with handcrafted priors do the job? I might also be wrong. Could you please elaborate?

-Qualitatively speaking it seems to me that the visual performances are not very good when compared to existing denoising algorithms (maybe a quantitative comparison in term of PSNR with other algorithms would be relevant, rather than only showing qualitative results). Also the multi step denoiser seems to give a lot of artefacts in figure 1.

-Aren't there missing reference in the related work regarding blind denoising ?

-p5 "the noise makes the support of the score function to be whole space, [...] non-Gaussian or off-the-manifold samples". I don't understand that part.


At this stage I give a weak accept, but I would consider raising my score if authors answer my concerns.

Typos:

p8 "without any re-trianing."

p7 "smoothingon"

p4 " matching obejctiv"

---

> ### Author Response · Authors · 2020-11-13
> **Response to AnnonReviewer 2 (1)**
>
> We sincerely appreciate your instructive review. We list the response for your questions below:
>
> Q1: If I understand well, one-step denoiser can only handle gaussian noise whereas multi-step can be applied to any log concave distribution? What is the advantage of one-step denoiser over multistep, does it perform better for gaussian noise?
> A1: Yes. The one-step denoiser is equivalent to a trained denoising autoencoder (DAE) which were used in [1]. DnCNN and MemNet are exemplar for such approach. On the other hand, the multi-step denoiser is a iterative algorithm that optimizes maximum a posteriori (MAP) loss with gradient approximated by score function, which allows various log-concave noise distributions.
>   The one-step denoiser is much faster than multi-step denoiser as it only requires one forward pass to achieve a denoised image. Moreover, the empirical results show that one-step denoiser performs better than multi-step denoiser for Gaussian test setting. On the other hand, multi-step denoiser is 'efficient' that can deal with various noise types.
>
> Q2: It is not clear to me why the method does not make use of the pretrained classifier but still outperforms Salman et al. which can acess the classifier? Is it only due to the denoiser's performance?
> A2: The performance gap between [1] and our can be explained with two aspects: the use of better network architecture and the multi-scale training. We used the U-net based architecture with improved convolution layer as illustrated in [2] which is developed for score-based modeling. Besides the impact of architecture, we empirically found out that multi-scale training slightly boosts the performance (Appendix C).
>  Moreover, even though we didn't include in the paper as we think is irrelevant to our interest, we trained score network with classification regularization as done in [1]. We trained the loss function as following:
> $$
> L_{\text{score}} + \gamma D_{KL}(f(x) \| f(\hat{x}))
> $$, where $\hat{x} = x -\sigma^2s_{\theta,\sigma}(\tilde{x})$ is denoised image from our one-step denoiser. We figure out that the classifier loss marginally improves the performance when $\sigma$ is large, but overall there were no difference.

---

> ### Author Response · Authors · 2020-11-13
> **Response to AnnonReviewer 2 (2)**
>
> Q3: I do not understand well the motivation for using score-based methods. Why scored based denoising in particular and not other blind denoising models? The current method is impacted by the size of the images, while most of the existing blind denoising algorithms are not affected by the size of the image. Actually, why even focus on learning base methods? Cannot simpler methods with handcrafted priors do the job? I might also be wrong. Could you please elaborate?
> Q5: Aren't there missing reference in the related work regarding blind denoising ?
>
> A3, A5: We aren't really certain for the definition of 'blind denoising', but if we assume blind denoising refers to denoising without the knowledge on the noise, then our framework is not a blind denoising, because we select the noise type and magnitude to decide the robustness. Moreover, our multi-step denoiser require the form of noise distribution.
>
> Note that non-learning image denoising algorithm such as BM3D can be applied to our denoised smoothing framework. However, many works shown that using deep image prior can boost the performance of image denoising. As [1] did, our method is built upon deep learning based image denoiser, and we generally divide into two types. First, training CNN based image denoiser such as DnCNN[3]. Those methods learns the feature to remove noise (especially Gaussian noise) from noisy images. Second, solving optimization with MAP formulation. Those method requires prior for image restoration, which the prior can be either handcrafted or by learning process. Many works showed that the generative model based approach is prominent, however those generative models such as GAN or Inveritible neural network took so long to denoise even when only small amount of noise is inserted. It is important to note that our score-based method covers both approaches with only one neural network.
> Moreover, we acknowledge that our algorithm degrades their performance when the dataset is of high dimension(e.g. ImageNet). The problem of high-dimensionality may alleviated by longer step of denoising, i.e. more minuscule sequence of $\sigma_i$. We suspect another approach such as striding with smaller patches to denoise, or downsampling and upsampling as done in [4]. We leave it for future work.
>
> Q4: Qualitatively speaking it seems to me that the visual performances are not very good when compared to existing denoising algorithms (maybe a quantitative comparison in term of PSNR with other algorithms would be relevant, rather than only showing qualitative results). Also the multi step denoiser seems to give a lot of artefacts in figure 1.
> A4: The visual performance might not be optimal compare to existing denoising algorithms. As mentioned above, We believe visiting more noise levels during MAP optimization processes may result in better denoising quality. However, due to the time bottleneck of randomized smoothing, we mediate the visual quality vs time tradeoff. It is definitely true that the better denoiser can perform better denoised smoothing but it really doesn't require ultimate clean and sharp image. The results from [1] supports that their classifier induced denoisers have strange artifacts but still perform well in denoised smoothing setting.

---

> ### Author Response · Authors · 2020-11-13
> **Response to AnnonReviewer 2 (3)**
>
> Q6: p5 "the noise makes the support of the score function to be whole space, [...] non-Gaussian or off-the-manifold samples". I don't understand that part.
> A6: As demonstrated in [2], addition of Gaussian noise makes the support of data density to be the whole space (as Gaussian noise is well-defined in whole area). Therefore, the noise fills the low-density region and it makes the denoising score matching objective more amenable. We will revise the statement to be more accurate one.
>
> Overall, we sincerely thanks for kind and detailed review which helped us a lot.
>
> [1] Salman, Hadi, et al. "Denoised Smoothing: A Provable Defense for Pretrained Classifiers." Advances in Neural Information Processing Systems 33 (2020).
>
> [2] Song, Yang, and Stefano Ermon. "Improved techniques for training score-based generative models." Advances in Neural Information Processing Systems 33 (2020).
>
> [3] Zhang, Kai, et al. "Beyond a gaussian denoiser: Residual learning of deep cnn for image denoising." IEEE Transactions on Image Processing 26.7 (2017): 3142-3155.
>
> [4] Block, Adam, et al. "Fast Mixing of Multi-Scale Langevin Dynamics underthe Manifold Hypothesis." arXiv preprint arXiv:2006.11166 (2020).

---

### Official Review · AnonReviewer2 · 2020-10-28

**Rating:** 3
**Confidence:** 5

**Review:**

Update: I have read the author's response and decided to keep my review, confidence, and score.

---

Summary: randomized smoothing is a method to construct provably robust classifiers via additive Gaussian noises on the input. The authors propose to learn score functions as a means to denoise the randomized image prior to a trained classification model. As the denoising + pre-trained classifier architecture is already proposed, the contribution is only limited to the choice of using a score function. The justification and realization of the method is limited for two main reasons. See below.

1. Efficiency: one of the most critical bottleneck of randomized smoothing methods is the slow prediction time. The score-function based generative / denoising models are known for their slow sampling time, so the proposed method undermines randomized smoothing in efficiency.

2. Many design choices in this paper is not well justified.

2-1) How good does the RHS of Eq. (12) approximates the gradient descent procedure?

2-2) Even if the true gradient descent can be executed, the bound in Eq. (13) seems very bad in high dimension, thus the smoothed classifier will not be accurate unless the pre-trained classifier is already robust in the local region.

2-3) Clearly using the same score function for multiple $\sigma$ is suboptimal. Although the authors mentioned this part as an advantage, but it is not clearly compared to existing methods. Would existing methods fail if they use the same denoising function for multiple $\sigma$?

---

> ### Author Response · Authors · 2020-11-13
> **Response to AnnonReviewer 3**
>
> We sincerely appreciate your instructive review. We list the response for your questions below:
>
> 1 - We realize that the slow prediction time is bottleneck for randomized smoothing, and it isn't efficient. However, the efficiency that we deal in this paper is about generation of smoothed classifier. We didn't change any algorithm on original randomized smoothing, therefore is independent to our work.
>  On the other hand, it is true that putting denoiser in front of the classifier slows down the prediction time. We empirically found out that it tooks 2-3 time (which is similar to that of [1]) for one-step denoiser and 4-10 time(depends on the $\sigma$) compare to original randomized smoothing (a.k.a white-box smoothing) which seems reasonable for prediction time. As ImageNet is of high-dimension the multiplicity makes prediction more slower, we leave accelerating the prediction time for future work.
>
> 2-1. For our multi-step approach, there are two approximations involved. First is approximation of data distribution $p$ with $p_\sigma$, and second one is approximation of $-\log p_\sigma(x)$ with score function $s_{\theta,\sigma}(x)$. Note that the first approximation error is bounded by $\sigma\sqrt{d}$ (which is shown in Appendix A) and second approximation error is generalization error which can be expressed by Radamacher complexity of used neural network for score network. We refer [2] for theoretic overview on learning score-based model (or equivalently DAE).
>
> 2-2 As we demonstrated in our theorem, the bound is loose when image is of high-dimension. However, as our denoising algorithm anneals $\sigma\rightarrow 0$, theoretically we can achieve small error bounds using extremely small $\sigma_L$. Furthermore, it is true that robust pre-trained classifier helps the performance. In fact, we experimented with classifier trained with very small Gaussian noise $\sigma = 0.01$, but it didn't help the performance. Instead, as illustrated in Appendix C.2 the classifiers with better test accuracy results in better performance.
>
> 2-3 We didn't really understand your question. But based on our interpretation, even though we use only one score function for multiple $\sigma$ the score function is conditioned by predetermined sequence of $\sigma$. Can you explain more about 'existing methods'? We hope your answer and willing to response.
>
> Overall, we sincerely thanks for kind and detailed review which helped us a lot.
>
> [1] Salman, Hadi, et al. "Denoised Smoothing: A Provable Defense for Pretrained Classifiers." Advances in Neural Information Processing Systems 33 (2020).
>
> [2] Block, Adam, Youssef Mroueh, and Alexander Rakhlin. "Generative Modeling with Denoising Auto-Encoders and Langevin Sampling." arXiv preprint arXiv:2002.00107 (2020).

---

> > ### Comment · AnonReviewer2 · 2020-11-13
> > **Response**
> >
> > 1 - The inefficiency of the proposed method and randomized smoothing are indeed independent. However, it raises a serious concern that the proposed method significantly exacerbate the weakness (inefficiency) in the original randomized smoothing framework.
> >
> > 2-1. I think my main concern is that your theorem does not really apply to the RHS of Eq. (12). You can of course combine results in the literature (similar to your response above) to get a theorem that applies to your setting, but this is not shown in the paper.
> >
> > 2-2. The argument is weak because you would never let $\sigma \to 0$. If $\sigma \to 0$, it means that randomized smoothing almost takes no effect.  I think $\sigma \sqrt{d}$ doesn't seem to be small even with $\sigma = 0.01$ in ImageNet scale datasets. I think your experiment results only highlight the fact that your theoretical analysis does not quite capture the empirical phenomenon.
> >
> > 2.3. Sorry for the confusion here. To clarify, using the same score function for multiple $\sigma$ is clearly a suboptimal choice. If you want to sell the point that your method works in that scenario, you have to highlight that your denoiser can work for multiple $\sigma$ while other methods do not work (e.g., the Denoised Smoothing paper). Even if that's the case, I still don't see why it is important to have a method that only needs one denoiser for multiple $\sigma$. Since the computational bottleneck of randomized smoothing is the prediction time rather than training time, why not just train one denoiser for each $\sigma$?

---

> > > ### Author Response · Authors · 2020-11-18
> > > **Response**
> > >
> > > Thank you for response.
> > >
> > > 1 - We admit that our framework may exacerbate the prediction time, but we claim that our method is efficient for generating the certified robust classifier.
> > >
> > > 2-1. We wil revise the theorem with explications with literature that we've mentioned.
> > >
> > > 2-2. To be clear, let $\sigma_R$ be that used for noise in randomized smoothing, then for denoising we use $\sigma_i$ such that
> > > $$
> > > \sigma_R > \sigma_1 > \sigma_2 \cdots > \sigma_L
> > > $$
> > > Therefore, by taking $\sigma_L$ as small as  possible, we can reduce the upper bound of the error. Therefore $\sigma_L\rightarrow 0$ is quite irrelevant to the effect of randomized smoothing.  We admit that $\sigma=0.01$ might be loose, but as result shown in CIFAR-10, the empirical results reflect the theorem.
> > >
> > > 2-3. Thank you for your explanation. Many image denoising literatures [1,2] pointed out that denoising autoencoder based image denoiser can't handle noise that differs to that used for training, which is one of the reason why the author of Denoised smoothing paper trained denoiser for each $\sigma$. To justify the motivation of our work, note that the $\sigma$ controls the robustness and accuracy tradeoff in randomized smoothing. Previous literatures that dealt with randomized smoothing used 3-4 $\sigma$s to evaluate the performance, also the denoised smoothing paper does. But for instance, the user of vision API might want to select their own $\sigma$ that fits to their samples. Then training new classifier or denoiser might be inefficient. However, our framework suggest that even without retraining the classifier or denoiser we can achieve certified robustness by using our denoiser.
> > >
> > > [1]. Guo, Shi, et al. "Toward convolutional blind denoising of real photographs." Proceedings of the IEEE Conference on Computer Vision and Pattern Recognition. 2019.
> > >
> > > [2]. Zhou, Yuqian, et al. "When awgn-based denoiser meets real noises." Proceedings of the AAAI Conference on Artificial Intelligence. Vol. 34. No. 07. 2020.

---

### Official Review · AnonReviewer3 · 2020-10-29
**Good work with some incremental novel contributions**

**Rating:** 6
**Confidence:** 3

**Review:**

This paper presents a denoising-based method for randomized smoothing that converts a base classifier into a smoothed one with p-robustness to adversarial examples. It considers a practical setting where the retraining/finetuning of the base classifier is largely inapplicable (e.g. the commercial classification service with only API provided to users).  To do this, it adopts a recently proposed methodology termed denoised smoothing [1] by prepending a custom-trained denoiser to the pretrained classifier. The major novelty of this work lies at the proposed denoising method using learned score function. The new denoising method only requires training one score network and is readily applicable to defend various $l_p$ adversaries, which is a key feature not available in [1].  The experiments show the proposed method outperforms the previous denoising-based approach, and is sometimes on par with the white-box approach [2] that manipulates the classifier.

Basically, this is an incremental work over [1] but the contributions claimed are perceived myself (though I have to admit I'm an expert on image denoising, instead of adversarial defense)
However, I do have some concerns about the method and the experiments, listed as follows:

- The major advantage of the score-function-based denoiser is the flexibility to handle various noise types and levels.  I don't expect it can beat the specialized Gaussian denoiser [1] under Gaussian perturbation setting.  As it is the case on Table 1/2, I'm wondering what's the benefit of  the proposed denoiser over the state-of-the-art Gaussian denoisers (as used in [1]) under Gaussian noise setting?
- The flexibility to tackle various $l_p$ adversary, the key feature of the proposed method is not thoroughly evaluated. In Table 2, I suggest the authors to add comparisons to [1] with denoiser trained on Gaussian noise setting, as well as ones trained with noise type aligned with the test setting.

[1] Denoised Smoothing: A Provable Defense for Pretrained Classifiers, Arxiv 2020
[2] Certified Adversarial Robustness via Randomized Smoothing, ICML 2019

---

> ### Author Response · Authors · 2020-11-13
> **Response to AnnonReviewer 3**
>
> We sincerely appreciate your instructive review. We list the response for your questions below:
>
> For clarification, [1] trained denoising autoencoder (DAE) with MSE and classification loss, and our one-step denoiser is equivalent to DAE trained with MSE loss. The equivalence of DAE and denoising score matching (DSM) is studied through various literatures [] as we demonstrated in the paper. Therefore the perforemance gap between [1] and our one-step denoiser is due to architecture change and multi-scale training. We used U-net based architecture as in [2], and we've shown that the multi-scale training benefits us learning the score function easier in Appendix C.
>
> Note that we trained only one score network which learns score of perturbed data distribution with Gaussian noise. For the best of our knowledge, [1] or any other works experimented with denoiser trained with other noise types such as Laplace noise or uniform noise. For your information, soon We will add comparison for Gaussian noise settings.
>
> Overall, we sincerely thanks for kind and detailed review which helped us a lot.
>
> [1] Block, Adam, Youssef Mroueh, and Alexander Rakhlin. "Generative Modeling with Denoising Auto-Encoders and Langevin Sampling." arXiv preprint arXiv:2002.00107 (2020).
>
> [2] Vincent, Pascal. "A connection between score matching and denoising autoencoders." Neural computation 23.7 (2011): 1661-1674.

---

> > ### Comment · AnonReviewer3 · 2020-11-13
> > **Thanks**
> >
> > Thanks for your response.  I'd like to see more results to address my concern on point #2

---

### Public Comment · ~Ruoxin_Chen1 · 2020-11-11
**Novelty**

Your idea is very similar with [1]. Both works propose a denoiser to construct robustness. Could you  compare your work with [1] and show your advantages.

[1] Salman, H., Sun, M., Yang, G., Kapoor, A., & Kolter, J. Z. (2020). Black-box smoothing: A provable defense for pretrained classifiers. arXiv preprint arXiv:2003.01908.

---

> ### Author Response · Authors · 2020-11-13
> **Response from authors**
>
> Thanks for your interest on our work.
>
> Our work can be seen as a generalization for [1]. They used CNN based image denoiser and proposed denoised smoothing. We built upon same denoised smoothing framework, but we devised score-based image denoiser which includes the method that used in [1]. Moreover, our work requires only one score network, while [1] trained multiple denoisers with respect to each noise levels. Therefore, our method is more efficient.
>
> [1]. Salman, Hadi, et al. "Denoised Smoothing: A Provable Defense for Pretrained Classifiers." Advances in Neural Information Processing Systems 33 (2020).

---

### Public Comment · ~Saeed_Saremi1 · 2020-11-16
**Missing the literature on seminal works and overlaps with existing work**

There are nice ideas in this submission but the authors do not seem to be aware of seminal works in empirical Bayes. Proposition 3.1 has been known for over 50 years. There are also significant overlaps both in concepts and in the methodology used in this paper and [SS20] as stated below:

(i) Proposition 3.1 goes back to [Rob56] which showed that the least-squares estimator of X (clean signals) given a measurement Y=y (noisy data) is the Bayes estimator xhat(y) = E[X|y] (for any noise model). For Gaussian noise, this Bayes estimator reduces to the expression in Eq. 5 (in the current paper) which has been known since [Miy61]. This literature has been reviewed extensively in [SH19] (Sec. 3).

(ii) We encourage the authors to take a look at [SS20] which starts with Empirical Bayes Smoothed Classifier (Definition 1.1) and further analysis in Proposition 2.3, essentially giving a full characterization of why the denoising should help in certification. But in addition it points out how it could fail. This later point is discussed in Sec 2.2, “Two effects of Bayes estimation” and led us to integrate the “score-based denoising” (as it is called in the current submission) with adversarial training. In particular, an algorithm called XHAT(epsilon) was presented for learning classifiers by combining empirical Bayes (akin to “score-based image denoiser”), randomized smoothing and adversarial training end-to-end.

(iii) Finally, the paper [XWM+19] is also very close to the spirit of this work and we think it needs to be at least mentioned.

References

[Rob56] Herbert Robbins. An empirical Bayes approach to statistics

[Miy61] Koichi Miyasawa. An empirical Bayes estimator of the mean of a normal population

[XWM+19] Cihang Xie, Yuxin Wu, Laurens van der Maaten, Alan L Yuille, and Kaiming He. Feature denoising for improving adversarial robustness, arXiv:1812.03411

[SH19] Saeed Saremi and Aapo Hyvarinen. Neural empirical Bayes. JMLR, 2019.

[SS20] Saeed Saremi and Rupesh Srivastava. Provable Robust Classification via Learned Smoothed Densities, arXiv:2005.04504, 2020.

---

> ### Author Response · Authors · 2020-11-24
> **Response**
>
> Thanks for your interest in our work. We appreciate your instructions and made some revisions to our rebuttal version.
>
> (1) We found out that the concept of empirical Bayes aligns with our proposition 3.1, and we added references to them.
>
> (2) Our work and the paper [SS20] seem to share a very similar idea, that they both use Bayes estimator for provable robust classification. Yet, we didn't conduct adversarial training since we only consider attaching denoiser to the pre-trained classifier, and we didn't experiment on the MNIST dataset. We inserted a short reference on [SS20] in section 2.2.
>
> (3) Since we consider provable methods for adversarial robust classification, we think [XWM+19] is irrelevant to our work. However, we appreciate your suggestion.
>
> [SS20] Saeed Saremi and Rupesh Srivastava. Provable Robust Classification via Learned Smoothed Densities, arXiv:2005.04504, 2020.
>
> [XWM+19] Cihang Xie, Yuxin Wu, Laurens van der Maaten, Alan L Yuille, and Kaiming He. Feature denoising for improving adversarial robustness, arXiv:1812.03411

---

### Author Response · Authors · 2020-11-24
**Revision uploaded**

We uploaded a revised version that reflected reviews and instructions from reviewers and commenters. We sincerely appreciate everyone's instruction.

---

### Decision · Program_Chairs · 2021-01-07
**Final Decision**

**Decision:**

Reject

**Comment:**

The paper proposes an improved method for randomized smoothing, reducing computationally complexity compared with some previous works. The authors propose to learn score functions to denoise the randomized image prior to feeding it to a trained classification model. More specifically,  two image denoising algorithms based on score estimation are proposed to be applied regardless of noise level/type.


Strengths:
- The paper shows strong quantitative results. The gap with white box smoothing is small on cifar, outperforming Salman et al. However according to the authors, the performance advantage could be mainly attributed to (1)  the use of better network architecture and (2) the multi-scale training, not the major contribution of a score-based denoiser.
- The denoiser doesn't require access to the pre-trained classifiers.
- The proposed method only requires training of one score network to handle various types of noise type/level, although reviewers have raised concerns about motivation to having a method that only needs one denoiser for multiple noise levels -  the computational bottleneck of randomized smoothing is the prediction time rather than training time and  using the same score function for multiple noise levels could be suboptimal.

Weaknesses:
- There are some concerns about the significance of the contribution as well as novelty of the work, as the denoising + pre-trained classifier architecture is already proposed. Specifically, the work can be seen as incremental to [1], although the work uses a score-based image denoiser whereas [1] uses a CNN based image denoiser and this work is more efficient as it requires only one score network, while [1] trained multiple denoisers with respect to each noise levels.
- Reviewers have expressed concerns on the prediction efficiency of score-function based generative / denoising models.  The proposed method might exacerbate the weakness of randomized smoothing (i.e., slow prediction), especially in high-dimensions.
-The reviewers are curious to see the benefit of the proposed denoiser over the state-of-the-art Gaussian denoisers (as used in [1]) under Gaussian noise setting.
-Method seems to be effective for low-resolution images only. The gap with white box increases on Imagenet.

[1]. Salman, Hadi, et al. "Denoised Smoothing: A Provable Defense for Pretrained Classifiers." Advances in Neural Information Processing Systems 33 (2020).